# HER2-Driven Breast Cancer: Role of the Chaperonin HSP90 in Modulating Response to Trastuzumab-Based Therapeutic Combinations

**DOI:** 10.3390/ijms26146593

**Published:** 2025-07-09

**Authors:** Italia Falcone, Elena Giontella, Stefano Giuliani, Giulia Borghesani, Alessandro Valenti, Valentina Zambonin, Sara Monteverdi, Luisa Carbognin, Emilio Bria, Ludovica Ciuffreda, Fabiana Conciatori, Chiara Bazzichetto, Serena Pedron, Alessia Nottegar, Sara Zanelli, Alice Muzzarelli, Alessandra Fabi, Silvia Migliaccio, Elisabetta Ferretti, Roberto Bei, Elena Fiorio, Maurizio Fanciulli, Isabella Sperduti, Anna Caliò, Michele Milella

**Affiliations:** 1SAFU, Department of Research, Advanced Diagnostics, and Technological Innovation, IRCCS-Regina Elena National Cancer Institute, 00144 Rome, Italy; italia.falcone@ifo.it (I.F.); stefano.giuliani@ifo.it (S.G.); ludovica.ciuffreda@ifo.it (L.C.); maurizio.fanciulli@ifo.it (M.F.); 2Section of Innovation Biomedicine—Oncology Area, Department of Engineering for Innovation Medicine, University of Verona and Verona University and Hospital Trust (AOUI), 37134 Verona, Italy; giontelena@gmail.com (E.G.); giulia.borghesani@gmail.com (G.B.); valentina.zambonin@aulss9.veneto.it (V.Z.); sara.monteverdi@apss.tn.it (S.M.); sara.zanelli@aovr.veneto.it (S.Z.); alice.muzzarelli@hotmail.it (A.M.); elena.fiorio@aovr.veneto.it (E.F.); michele.milella@univr.it (M.M.); 3Radiology and Diagnostic Imaging Unit, Department of Clinical and Dermatological Research, San Gallicano Dermatological Institute IRCCS, 00144 Rome, Italy; alessandro.valenti@ifo.it; 4UOC Oncologia Medica, Comprehensive Cancer Center, Fondazione Policlinico Universitario Agostino Gemelli IRCCS, 00186 Roma, Italy; luisa.carbognin@gmail.com (L.C.); emilio.bria@unicatt.it (E.B.); 5Medical Oncology, Department of Translational Medicine and Surgery, Università Cattolica del Sacro Cuore, 00168 Roma, Italy; 6Medical Oncology Unit, Ospedale Isola Tiberina-Gemelli Isola, 00186 Rome, Italy; 7Preclinical Models and New Therapeutic Agents Unit, IRCCS-Regina Elena National Cancer Institute, 00144 Rome, Italy; fabiana.conciatori@ifo.it (F.C.); chiara.bazzichetto@ifo.it (C.B.); 8Department of Diagnostic and Public Health, Section of Pathology, University of Verona, 37134 Verona, Italy; serena.pedron@univr.it (S.P.); alessia.nottegar@univr.it (A.N.); anna.calio@univr.it (A.C.); 9Precision Medicine in Breast Cancer Unit, Scientific Directorate, Department of Woman and Child Health and Public Health, Fondazione Policlinico Universitario Agostino Gemelli IRCCS, 00168 Rome, Italy; alessandra.fabi@policlinicogemelli.it; 10Department of Experimental Medicine, University of Rome “Sapienza”, 00161 Rome, Italy; silvia.migliaccio@uniroma1.it (S.M.); elisabetta.ferretti@uniroma1.it (E.F.); 11Department of Clinical Sciences and Translational Medicine, University of Rome “Tor Vergata”, 00133 Rome, Italy; bei@med.uniroma2.it; 12Biostatistics Unit, IRCCS-Regina Elena National Cancer Institute, 00144 Rome, Italy

**Keywords:** HSP90, HER2, breast cancer, combination treatments, predictive biomarker, therapeutic strategies

## Abstract

Mechanistic relationships between heat shock protein 90 (HSP90) and human epidermal growth factor receptor 2 (HER2) are complex and clinical correlations in breast cancer remain inconsistent. We investigated the role of HSP90 expression in the response of breast cancer cells to HER2-targeted treatments, by measuring cell viability/proliferation and protein expression after genetic and pharmacologic HER2/HSP90 modulation. HSP90 expression was also assessed by immunohistochemistry in a series of 72 metastatic, HER2+ breast cancer patients. In HER2+ breast cancer models (AU565, BT474, MCF7-HER2), HER2 downregulation induced HSP90 upregulation and growth inhibitory synergism between trastuzumab and docetaxel. HSP90 downregulation blunted the response to trastuzumab and docetaxel and their synergistic interactions. The addition of pertuzumab caused little additional growth inhibition, but HSP90 silencing unmasked a synergistic growth inhibitory effect with the triple combination. Conversely, HSP90 downregulation blunted the therapeutic response to trastuzumab/pertuzumab/tamoxifen or trastuzumab–emtansine. In HER2+ breast cancer patients, high HSP90 expression was associated with significant progression-free survival benefit with the triple combination, as compared with trastuzumab and chemotherapy, although the interaction test was not statistically significant. Overall, our results highlight a mechanistic role for HSP90 in determining the response of breast cancer cells to HER2-targeted agents and suggest that trastuzumab/pertuzumab combinations may be particularly advantageous in HSP90-high, HER2+ breast cancer.

## 1. Introduction

Approximately 15–20% of breast cancers are characterized by amplification and overexpression of the human epidermal growth factor receptor 2 (HER2, also known as. ErbB2), a member of the ErbB family, being able to homo- or heterodimerize with other family members (HER1/HER3/HER4) in a ligand-independent fashion [1]. Downstream signaling activated by HER2 dimerization and kinase activation play vital roles in breast tumorigenesis, resulting in aggressive biological and clinical behavior, and is associated with resistance to conventional chemotherapeutic and hormonal treatments and poor prognosis [2].

The introduction of trastuzumab, a humanized monoclonal antibody directed against the extracellular subdomain IV of the HER2 protein, has paved the way for the development of HER2-targeted therapies in HER2-overexpressing (hereafter referred to as HER2+) cancers, particularly breast cancer [3]. In HER2+ breast cancer, the array of therapeutic weapons nowadays encompasses trastuzumab alone combined with taxane-based chemotherapy and/or hormone therapy in a low-risk adjuvant setting; the combination of trastuzumab and pertuzumab (a humanized monoclonal antibody directed against the HER2 dimerization domain) in combination with taxane-based chemotherapy and/or hormone therapy in the neoadjuvant, high-risk adjuvant, and first-line metastatic settings; and trastuzumab-based antibody–drug conjugates (ADCs), such as trastuzumab emtansine (T-DM1) and trastuzumab deruxtecan (T-Dxd), and/or HER2-targeted tyrosine kinase inhibitors (TKIs), such as lapatinib and tucatinib, after failure of trastuzumab/pertuzumab-based combinations [4]. Despite the undisputed role of HER2-targeted therapy in HER2+ breast cancer, response to agents targeting HER2 with different molecular mechanisms displays intra- and inter-patient heterogeneity [5,6]. This suggests that, besides HER2 expression, other factors may modulate sensitivity/resistance to HER2-targeted agents, specifically depending on their molecular mechanism of action, thereby offering opportunities for the development of predictive biomarkers and new therapeutic targets.

Heat shock protein 90 (HSP90) is a chaperonin involved in the maturation and proteolitic turnover of several client proteins, including HER2 [7,8]. Many of the proteins controlled by HSP90 are implicated in important cellular processes and HSP90 itself is often dysregulated in cancer cells; therefore, it is considered a potential biomarker and therapeutic target in cancer, against which different inhibitors are being developed [9,10,11]. In HER2+ breast cancer, the relationships between HSP90 and HER2 dimerization, trafficking, and degradation under physiological conditions or in response to trastuzumab or other HER2-targeting agents is complex and incompletely understood. Meanwhile, HSP90 stabilizes HER2 at the plasma membrane, thereby promoting its signaling ability and potentially resulting in poor outcomes and decreased survival [12,13,14,15], with the outcome of mechanistic interference with HSP90 expression and function, as well as clinical data in the HER2+ context, remaining inconsistent [13,16].

In this study, we investigated the role played by HSP90 in the response to HER2-targeted combination therapy in preclinical breast cancer models in vitro and explored the prognostic/predictive impact of HSP90 expression in a mono-institutional series of HER2+ breast cancer patients who received trastuzumab- or trastuzumab/pertuzumab-based combination therapy as their first-line treatment for metastatic disease.

## 2. Results

### 2.1. Relationships Between HER-2 and HSP90 Expression in Breast Cancer Cell Lines

We first analyzed HER2 and HSP90 protein expression by Western blot (WB) in a panel of 10 breast cancer cell lines with or without HER2 gene amplification (Appendix A) [17,18,19]. As reported in Figure 1A, no significant correlations between the basal expression of the two proteins were observed. However, HER2 silencing by stable RNA interference in the HER2-amplified AU565 cell line induced HSP90 upregulation, in parallel to HER2 downregulation. Conversely, stable transfection of HER2 in the HER2-negative MCF7 breast cancer cell lines did not result in HSP90 expression changes (Figure 1B). Moreover, exposure of HER2+ (AU565, BT474, and MCF7-HER2), but not HER2-negative [MCF7, ZR75-1 and MCF7-empty vector (MCF7-EV)], breast cancer cell lines to the anti-HER2 mAb trastuzumab for 72 h induced parallel downregulation of HER2 and upregulation of HSP90 proteins, respectively, as assessed by WB and immunofluorescence (Figure 1C,D). Conversely, exposure to the microtubule disruptor docetaxel for 24 h did not modulate HSP90 protein expression in any of the contexts analyzed (Figure 1E).

### 2.2. Synergistic Effect of the Combination of Trastuzumab and Docetaxel in HER2-Driven Breast Cancer Cell Lines

We next assessed the functional response of HER2+ and negative breast cancer cell lines to trastuzumab and docetaxel. As shown in Appendix A, dose–response curves showed selective cell growth inhibition in HER2+ cell lines (AU565 and BT474) upon exposure to single-agent trastuzumab, as well as greater growth inhibition in response to single-agent docetaxel in HER2+, as compared to HER2-negative (MCF7 and ZR75-1), breast cancer cell lines (Appendix A). As expected, the combination of trastuzumab and docetaxel resulted in significant growth inhibition only in HER2+ breast cancer cells, whereas no effect of trastuzumab, either alone or combined with docetaxel, was observed in HER2-negative cells (Figure 2A).

These results were further confirmed in MCF7-HER2 cells, which were sensitive to single-agent trastuzumab and more sensitive to single-agent docetaxel, as compared to MCF7-EV (Appendix A); moreover, growth inhibition with the combination of trastuzumab and docetaxel was selectively observed in MCF7-HER2 (Appendix A). From a molecular point of view, the combination of trastuzumab and docetaxel downregulated HER2 and upregulated HSP90 protein expression selectively in HER2+ breast cancer cell lines (AU565, BT474 and MCF7-HER2) after 72h of treatment (Figure 2B and Appendix A).

### 2.3. HSP90 Modulation Influences Cellular Response to HER2-Targeted Treatment in HER2-Driven Breast Cancer Cells

To confirm HSP90’s role in response to combination treatment, we silenced its expression by transient siRNA transfection in HER2+ (AU565, BT474, and MCF7-HER2) breast cancer cell lines. In this context, HSP90 downregulation significantly attenuated the growth inhibitory response of HER2+ breast cancer cells to both single-agent trastuzumab and docetaxel and their combination (Figure 3A,B and Appendix A).

These results were confirmed by stable HSP90 silencing by shRNA transfection; although modulation of HSP90 expression was less evident in this experimental setting, HSP90 silencing significantly decreased growth inhibition induced by both single-agent trastuzumab and docetaxel as well as their combination (Appendix A). Similar results were obtained by the pharmacological inhibition of HSP90 activity using geldanamycin; although hampered by the strong growth inhibitory effect of geldanamycin alone, pretreatment of HER2+ breast cancer cells with geldanamycin for 24 h significantly impaired further growth inhibition induced by single-agent trastuzumab and docetaxel treatments, as well as growth inhibitory synergism of their combination (Appendix A).

### 2.4. HSP90 Modulation Influences Cellular Response of HER2-Driven Breast Cancer Cells to the Combination of Trastuzumab, Pertuzumab, and Docetaxel

The combination of trastuzumab, pertuzumab, and docetaxel is the current standard of treatment for HER2+ breast cancer patients [20]. Therefore, we also investigated the potential role of HSP90 in the response of HER2+ breast cancer cell lines to such a combination. In the AU565 and BT474 models in vitro, pertuzumab, either alone or in combination with trastuzumab (Figure 4A,B) or docetaxel (Appendix A), caused little (if any) additional growth inhibition; in this context, HSP90 silencing by siRNA completely blocked the growth inhibitory effects of single-agent trastuzumab and pertuzumab, thereby unmasking a slight, but significant, increase in growth inhibition by the combination of trastuzumab and pertuzumab (Figure 4A,B). Consistently, HSP90 silencing by siRNA significantly blunted the response to single-agent docetaxel, thereby unmasking a growth inhibitory effect with the triple combination of trastuzumab, pertuzumab, and docetaxel (Figure 4C,D).

### 2.5. HSP90 Modulation Influences Cellular Response of HER2-Driven Breast Cancer Cells to the Combination of Trastuzumab, Pertuzumab, and Tamoxifen and Trastuzumab Emtansine (TDM-1)

In HER2+/estrogen receptor (ER)-positive breast cancer, maintenance treatment with trastuzumab, pertuzumab, and hormonal therapy is usually given after a fixed number of cycles of trastuzumab, pertuzumab, and docetaxel induction [21]. Moreover, ADC directed against HER2 (T-DM1 or trastuzumab/deruxtecan) currently represents the standard second-line treatment for advanced HER2+ breast cancer. We therefore investigated the effects of HSP90 modulation in response to the combination of trastuzumab, pertuzumab, and 4-OH tamoxifen in the BT474 HER2+/ER-positive breast cancer cell line. In this context, HSP90 silencing by siRNA significantly blunted the growth inhibitory response to tamoxifen alone and completely inhibited the anti-tumor effect of the combination observed in parental BT474 cells (Figure 5A). On the contrary, the growth inhibitory effect of T-DM1 in the AU565 model was significantly potentiated by HSP90 silencing by siRNA (Figure 5B and Appendix A).

### 2.6. Clinical Impact of HSP90 Expression in Advanced HER2+ Breast Cancer

We next retrospectively examined the impact of HSP90 expression on progression-free survival and overall survival (PFS and OS) of 72 patients with HER2+ breast cancer undergoing first-line treatment for metastatic disease with either trastuzumab plus chemotherapy or combined trastuzumab/pertuzumab plus chemotherapy. Median age at the time of diagnosis of metastatic disease was 60 years (range 37–84). A total of 24 patients (33%) had metastatic disease at diagnosis, 42 (58%) and 30 (42%) received chemotherapy with trastuzumab alone or the combination of trastuzumab and pertuzumab, respectively, and 42 (58%) patients also received maintenance hormonal therapy in combination with either trastuzumab or the combination of trastuzumab and pertuzumab. Patients’ characteristics are summarized in Table 1. In the population analyzed, median PFS from the start of first-line treatment was 26 months (CI95% 9.5–42.5 months) and median OS was 76 months (CI95% 49.1–102.8 months).

HSP90 protein expression was detected in all but one sample, with a 3+ staining intensity in at least a fraction of the tumor cells in 64 cases and positive staining in 100% of the neoplastic cells in 63. Using a hybrid variable, obtained by multiplying staining intensity (score 0–3) by the percentage of positive cells (0–100), the distribution of HSP90 expression relative to the clinical outcome of PFS was preliminarily conducted, identifying a cut-off value of 250 to divide the cases into high (≥250) and low (<250) HSP90 expression (Appendix A). Using this cut-off, no statistically significant differences in either PFS or OS from the start of first-line treatment were observed for high or low HSP90 expressors (Appendix A); however, a non-significant (*p* = 0.057) trend towards better PFS from the start of maintenance hormonal treatment (HT) was observed for patients whose tumors expressed low HSP90 levels (Appendix A). Similarly, HSP90 expression did not impact survival outcomes (PFS and OS from the start of first-line treatment) when patients who received trastuzumab and chemotherapy or combined trastuzumab/pertuzumab and chemotherapy were analyzed separately (Appendix A); however, among HER2+/ER-positive patients who received maintenance HT, PFS was significantly better in low HSP90 expressors who had received trastuzumab monotherapy induction and maintenance (*p* = 0.024; Appendix A).

As expected from available evidence from randomized trials [20], induction and maintenance treatment with combined trastuzumab/pertuzumab resulted in significantly longer PFS and OS, as compared with trastuzumab-only induction and maintenance, in the population analyzed (Appendix A). When analyzed separately for low and high HSP90 expressors, a significant PFS improvement by the combination of trastuzumab and pertuzumab, as compared with trastuzumab monotherapy, was only observed for patients expressing high HSP90 levels (*p* = 0.006; Figure 6A); such an advantage in PFS for combined trastuzumab/pertuzumab induction appeared to be carried over to a significant improvement in PFS to maintenance HT for high HSP90 expressors who received combined trastuzumab/pertuzumab, as compared to trastuzumab monotherapy (*p* = 0.023; Appendix A), while only a non-significant trend towards PFS improvement was observed in low HSP90 expressors (*p* = 0.057).

However, a significant interaction between treatment and HSP90 expression was not detected (interaction *p* = 0.640 and 0.506 for PFS to induction and maintenance HT, respectively) and did not translate into a significant OS advantage for combined trastuzumab/pertuzumab, as compared to trastuzumab monotherapy, in high HSP90 expressors (Figure 6B). Accordingly, only baseline performance status and first-line biological therapy were found to be independent predictors of PFS at multivariate analysis (Table 2), while only performance status at the start of first-line treatment significantly impacted OS (Appendix A).

## 3. Discussion

Here, we investigated the role of HSP90 expression in the response of breast cancer cells to HER2-targeted treatments, using both preclinical cell line models and a mono-institutional series of metastatic, HER2+ breast cancer patients who underwent trastuzumab- or trastuzumab/pertuzumab-based first-line treatment. Although no correlation was found between HER2 and HSP90 expressions in the panel of breast cancer cell lines examined, HER2 downregulation resulted in compensatory upregulation of HSP90 expression. From a functional standpoint, HSP90 downregulation significantly blunted the growth inhibitory response of HER2+ breast cancer cell lines to single-agent and combined trastuzumab and docetaxel and abrogated their therapeutic cooperation. In the same cell line models, pertuzumab in combination with trastuzumab or docetaxel caused little (if any) additional growth inhibition; in this context, HSP90 silencing completely blocked the growth inhibitory effects of single-agent trastuzumab, pertuzumab, and docetaxel, thereby unmasking a slight cooperative, growth inhibitory effect with their triple combination. Conversely, cooperative interactions between combined trastuzumab/pertuzumab and tamoxifen, as well as single-agent activity of the anti-HER2 ADC T-DM1, were significantly impaired by HSP90 downregulation. Finally, in HER2+, metastatic, breast cancer patients, high HSP90 expression did not have prognostic impact in terms of PFS or OS. Although, as expected, the combination of trastuzumab and pertuzumab achieved significant and borderline significant prolongation of PFS and OS, respectively, in the overall population, such an effect was particularly pronounced in the HSP90-high population. In HER2+/ER-positive patients undergoing hormonal maintenance, patients with high HSP90 expression displayed a non-significant trend towards worse PFS, particularly in the subset of patients who received trastuzumab monotherapy. Indeed, in this patient subpopulation, the use of a combination of trastuzumab and pertuzumab in the induction and maintenance phase achieved a borderline significant and significant prolongation of PFS from the start of hormonal maintenance, as compared to single-agent trastuzumab, in HSP90-low and -high patients, respectively.

High HSP90 expression in breast cancer cell lines, independent of their HER2 status, has been previously reported [13]; in clinical series, HSP90 expression displays more variability and a relatively inconsistent association with HER2 overexpression [13]. Since the clinical series reported here included only patients with HER2+ breast cancer, it cannot contribute to establishing an association between HER2 and HSP90 expression. However, in this series, HSP90 was highly expressed in the vast majority of patients (>85%). Moreover, downregulation of HER2 expression by genetic means or trastuzumab treatment in HER2+ cell lines induced HSP90 upregulation, thereby establishing a mechanistic relationship between HER2 and HSP90 expression. Although our data may seem to be in contrast with a previous report linking HER2 inhibition with downregulation of HSP90 expression [22], it has to be noted that the latter results were obtained using small-molecule HER2 tyrosine kinase inhibitors (CP724.714 and lapatinib), which act in a mechanistically distinct fashion, as compared to downregulation of HER2 protein expression [23]. HSP90 expression, which was not modulated by single-agent docetaxel, was also upregulated (in HER2+ breast cancer models) with the combination of trastuzumab and docetaxel under conditions that induced the expected growth inhibitory synergism [24], further strengthening the observed relationship between HER2 and HSP90 expression and its potential therapeutic relevance.

The dynamics of molecular interactions between HSP90 and its client protein HER2 are complex, and their functional relevance is still incompletely elucidated [25]. While the formation of HER2:HSP90 complexes is involved in the maturation, maintenance of the appropriate conformation, stabilization, and activation of HER2 [26], the functional effects of HSP90 inhibition (by either pharmacological intervention or protein downregulation) on HER2 dynamics and signaling, particularly in response to agents such as trastuzumab, are more controversial. Several reports have shown that pharmacological inhibition of HSP90 activity restores trastuzumab sensitivity in preclinical models of HER2+ breast cancer (reviewed in [27]); however, clinical trials with HSP90 inhibitors have met with little success so far [28] due to several reasons, including off-target effects and poor solubility [16,28]. On the other hand, another set of data available in the literature suggests that binding of HSP90 to the N-lobe of HER2 [29] may sequester HER2 homodimers, restrain catalytic activity, and prevent heterodimer formation [26,30,31], resulting in decreased HER2 activation and signaling; this hypothesis is supported by the finding that HER2 mutants that cannot bind HSP90 display increased kinase activity [32].

The set of experiments presented here was designed to assess the functional relevance of HSP90 expression in the response to HER2-targeted therapy, rather than HER2 dynamics and signaling in relationship to HSP90 modulation. In that respect, our data cannot contribute to settling the question of whether high HSP90 expression fosters or restrains HER2 signaling. From a functional standpoint, however, HSP90 downregulation by genetic means clearly blunts the response to combined trastuzumab/docetaxel, thereby abrogating their therapeutic synergism. These results are consistent with clinical data previously reported by our group, suggesting that HSP90-overexpressing breast cancer clones may be more effectively eliminated by trastuzumab–docetaxel-based treatment, thereby translating into a higher proportion of pathologic complete response (pCR) [33]. Such a hypothesis is indeed supported by evidence that residual cells in tumors that do not achieve pCR displayed a lower HSP90 immunostaining, thereby mirroring the lack of treatment efficacy on residual tumor cell clones, similar to what is observed for Ki67 [34,35,36]. Although these findings may seem in contrast with previous evidence suggesting that HSP90 inhibition synergizes with trastuzumab in preclinical models of HER2+ breast cancer [37], it has to be noted that such evidence employed pharmacological HSP90 inhibition [37,38,39,40], rather than modulation of its expression, and mostly demonstrated activity of such inhibitors, alone or in combination with trastuzumab, in HER2+ models that had already become resistant to trastuzumab [38]. Unfortunately, data obtained with geldanamycin in the present report, suggesting little or no additional effect of combined trastuzumab/docetaxel, are difficult to interpret due to the pronounced growth inhibitory effects of single-agent HSP90 inhibition.

Much less is known regarding the molecular mechanisms of potential interactions between HSP90 and other HER2-targeted treatments currently in use, such as pertuzumab and T-DM1. Our results, showing that HSP90 downregulation unleashes growth inhibitory cooperation with the triple combination of trastuzumab/pertuzumab and docetaxel and is sensitive to the growth inhibitory effects of T-DM1, are consistent with a proposed mechanism by which HSP90 inhibition may cooperate with pertuzumab in favoring HER2 internalization and degradation [41] and, therefore, with T-DM1 activity (reviewed in [41,42,43,44,45]). On the other hand, our data show complete loss of therapeutic cooperation between trastuzumab/pertuzumab and tamoxifen upon HSP90 downregulation in cellular models of HER2+/ER-positive breast cancer. These results are consistent with published preclinical data showing that 4-OH-tamoxifen (the active metabolite of tamoxifen) may act as a HSP90 activator by stimulating its ATPase activity [46], and that modest HSP90 inhibition, below the threshold for proteotoxic activation of the heat shock response, dramatically impairs the emergence of tamoxifen resistance in cell culture and mouse models of breast cancer [47], raising the interesting hypothesis that an intact HSP90 activity is indeed necessary to take full advantage of tamoxifen-based combinations in this context.

How do we translate preclinical findings into the clinical scenario of metastatic HER2+ breast cancer? Results obtained in our series, although limited, are in line with the published literature and current guidelines with regard to the greater efficacy of the triple combination of pertuzumab/trastuzumab and docetaxel, as compared with trastuzumab and docetaxel [20]. In this context, no significant association was observed between HSP90 expression and PFS/OS from the start of first-line treatment. The potential prognostic role of HSP90 in human malignancies is ambiguous and appears to depend on the specific tumor type considered [48,49]. Our findings are in line with inconsistent associations between survival outcomes and HSP90 expression in different breast cancer subtypes [50]. Indeed, in early-stage disease, studies by Jameel et al. [51] and Cheng et al. [52] found an association of borderline significance between worse DFS/OS and high HSP90 expression (synthesized in [50]); however, the latter study showed that these results pertained only to the triple-negative subgroup of patients and no association could be established in the HER2+ or HER2-negative/ER+ subgroups [52]. Interestingly, although a significant interaction between HSP90 and the type of anti-HER2 treatment (trastuzumab/pertuzumab vs. trastuzumab alone) could not be demonstrated, the increased efficacy of trastuzumab/pertuzumab combinations in terms of PFS appears to be mostly confined to the HSP90-high subgroup of patients. These results were consistent with those obtained in the hormonal maintenance setting in HER2+/ER-positive patients: (1) patients treated with trastuzumab-only induction/maintenance (but not those receiving trastuzumab/pertuzumab-based induction/maintenance) fared significantly better upon hormonal maintenance if they had low HSP90 expression; (2) although a significant interaction between HSP90 and the type of anti-HER2 treatment (trastuzumab/pertuzumab vs. trastuzumab alone) could not be demonstrated, the increased efficacy of trastuzumab/pertuzumab combinations, in terms of PFS, upon hormonal maintenance appears to be particularly striking (and statistically significant) in patients with high HSP90 expression. These latter results are consistent with the lack of therapeutic cooperation between trastuzumab/pertuzumab and tamoxifen upon HSP90 downregulation observed in our in vitro models as well as with the published literature, supporting the idea that an HSP90 activity is indeed necessary to take full advantage of tamoxifen growth inhibitory activity (see above [46,47]).

## 4. Materials and Methods

### 4.1. Cell Lines

All cell lines of this study were obtained from the American Type Culture Collection (ATCC) (Appendix A) [17,18,19]. Cell lines were routinely maintained in Dulbecco’s Modified Eagle’s Medium (DMEM) supplemented with 10% fetal bovine serum (FBS), 2 mM L glutamine, and antibiotics (Pen/Strep) in a humidified atmosphere with 5% CO_2_ at 37 °C. Cell culture reagents were purchased from Euroclone (Milan, Italy).

### 4.2. Western Blot Analysis

To obtain whole protein extract, cells were lysed in SDS lysis buffer containing 20 mmol/L Tris-HCl (pH 8.0), 100 mmol/L NaF, 1 mmol/L NaVO_4_, 10 mmol/L PMSF, 10 μg/mL leupeptin, 2% SDS. Protein concentration was detected by Bio-Rad Protein Assay Dye Reagent Concentrate (Bio-Rad, Hercules, CA, USA). Approximately 20 ng of total proteins were fractionated by SDS-polyacrylamide gel electrophoresis and transferred to the nitrocellulose membrane (Amersham, Arlington Heights, IL, USA). Membranes were probed with specific primary antibodies for total HER2 (Cell Signaling Technology Inc., Beverly, MA, USA Cat#2248; RRID: AB2099242) and HSP90 (Cell Signaling Technology Inc., Beverly, MA, USA Cat#4875; RRID: AB_2233331). To control the amount of proteins transferred to the nitrocellulose membrane, GAPDH (Cell Signaling Technology Inc., Beverly, MA, USA Cat#5174; RRID: AB_10622025) and β-Actin (Sigma-Aldrich, St. Louis, MO, USA Cat#A1978; RRID: AB_476692) were used. Membranes were incubated with peroxidase-conjugated anti-mouse or anti-rabbit secondary antibodies (Jackson Immunoresearch Labs, Inc., Baltimore, MD, USA) and developed with a chemi-luminescence (ECL) system (Amersham). For image detection, UVITEC Alliance 4.7 system (Cambridge, UK) was used.

### 4.3. Drug Treatments and Cell Proliferation Assay

For in vitro experiments, trastuzumab, pertuzumab, trastuzumab–emtansine (T-DM1), and docetaxel were provided by IFO Hospital pharmacy; tamoxifen and geldanamycin were acquired by MedChemExpress (Monmouth Junction, NJ, USA). Trastuzumab and pertuzumab and T-DM1 were dissolved in sterile water at concentrations of 21 and 30 mg/mL, respectively, and stored at −20 °C. Docetaxel (20 mg/mL) was stored at 4 °C. Tamoxifen and geldanamycin were dissolved in sterile Dimethyl Sulfoxide (DMSO) and stored at −20 °C. The final concentration of all drugs was obtained by dilution with culture medium. Cell proliferation was evaluated by Crystal Violet assay. Tumor cells were plated into 24-wells (Corning, Inc., Corning, NY, USA) at a concentration of 3 × 10^4^ cells/well. The following day, the drugs were added at indicated concentrations. After 72 h of treatment, the cells were washed three times with phosphate-buffered saline (PBS), fixed with 4% (vol/vol) formaldehyde, and stained with 0.1% crystal violet (CV) for 40 min at room temperature (RT). Excess stains were removed with water and the plates were dried. CV stain was extracted with 95% (vol/vol) acetic acid and the absorbance was measured at 570 nm.

### 4.4. RNA Transfection

Cells were transfected with a short interfering RNA (siRNA) targeting human HSP90α/β (sc-35608, Santa Cruz Biotechnology, Dallas, TX, USA). RNA was transfected with RNAiMax reagent (Invitrogen, Carlsbad, CA, USA) for 72 h according to manufacturer’s protocol. HSP90 downregulation was evaluated by Western blot analysis.

### 4.5. Plasmid Transfection

Cells were transfected with short hairpin RNA (shRNA) plasmids against HER2 (SABiosciences, Frederick, MD, USA) and HSP90 (Origene Technologies, Rockville, MD, UDA). Transfections were performed using the Lipofectamine 2000 transfection reagent (Invitrogen, Carlsbad, CA, USA) according to the manufacturer’s protocol. HER2 and HSP90 downregulation were evaluated by Western blot analysis.

### 4.6. Immunofluorescence

AU565, BT474, MCF7, and ZR75-1 cells were plated on coverslips and treated with trastuzumab for 72 h. After treatment, cells were fixed in 4% paraformaldehyde in PBS for 10 min, permeabilized in PBS containing 0.5% Triton X-100 for 10 min, and blocked with 5% BSA, 0.3% Triton X-100 in PBS for 90 min. Coverslips were incubated overnight with HSP90 primary antibody (Cell Signaling Technology Inc., Beverly, MA, USA Cat#4875; RRID:AB_2233331) diluted in PBS solution containing 1% BSA, 0.3% Triton X-100. The following day, coverslips were washed three times in PBS and incubated for 1 h at a dilution of 1:400 with goat anti-rabbit Alexa Fluor 488 (Thermo Fisher Scientific Cat# A11034, Waltham, MA, USA, RRID:AB_2576217) diluted in the same solution of the primary antibody. After three washes with PBS, the nuclei were counterstained with DAPI and the cells were observed under Zeiss Axiovert 200 M fluorescence microscope. Specific fields were photographed with a digital camera equipped with Zeiss Axiovision acquisition software 4.7.

### 4.7. Immunohistochemical Evaluation of HSP90 Expression

Formalin-fixed, paraffin-embedded (FFPE) tissue samples were collected and analyzed from the archives of the Pathology Unit of AOUI Verona. The histological material analyzed included the following: biopsy of the primary tumor, surgical specimen containing the neoplasm (when available), biopsy of metastatic sites. Whenever possible, the most recent tumor sample was analyzed. For each case, histology was reviewed under light microscopy to select the best block for immunohistochemical analysis with the HSP90 antibody (clone EPR16621-67, dilution 1:1500, Abcam Cat# ab223467, Cambridge, UK; RRID: AB_2943491). The immunohistochemical analysis was performed using the “Bond Polymer Refine” detection system on the automated Bond immunostainer (Leica Biosystems, Nussloch, Germany). For each sample, the percentage of positive neoplastic cells and staining intensity were recorded: 1 (weak), 2 (moderate), 3 (strong). Appropriate positive and negative controls were included in each run.

### 4.8. Study Population

Patients aged 18 years or older diagnosed with stage IV HER2-positive breast cancer, regardless of the site of metastasis, were included in this study. A total of 72 patients were retrospectively identified through records of trastuzumab and pertuzumab administration and were followed from 2005 to 2019 at the Oncology Department of the Azienda Ospedaliera Universitaria Integrata in Verona. Clinical records were reviewed to collect the following information: patient characteristics, including date of diagnosis, age at diagnosis, and ECOG performance status both at diagnosis and at the initiation of treatment for metastatic disease; treatment details, including start and end dates of first, second, third, and subsequent lines of therapy; disease characteristics, such as stage at diagnosis, tumor histotype, and biomolecular features (estrogen receptor (ER), progesteron receptor (PgR), Ki67, HER2 assessed by immunohistochemistry (IHC) and Fluorescence In Situ Hybridization (FISH)), site of the primary tumor, as well as the site and number of visceral metastases; and any surgery or radiotherapy performed on the primary tumor or metastatic sites (radical or non-radical). Outcome data collected included type of response (complete response, partial response, stable disease, or disease progression), date of radiological progression according to RECIST criteria based on whole-body imaging (e.g., computer tomography (CT), Positron Emission Tomography (PET), bone scan); in cases without documented progression, PFS was calculated from the start of treatment until the date of the last visit or therapy infusion. Additionally, OS was recorded based on the date of death or last follow-up, along with the start and end dates of each therapy line.

### 4.9. Statistical Analysis

Data are summarized using basic descriptive statistics. The Chi-square test or Fisher exact test were used for categorical variables, when appropriate. Differences between the groups were analyzed with a 2-tailed Student’s *t* test for paired samples. Hazard ratios (HRs) with 95% confidence intervals (95% CIs) were estimated for each variable of interest. A *p* value < 0.05 was considered significant. Only variables significant in univariate were included in the multivariate analysis. A multivariate logistic regression and proportional hazard model were developed using stepwise regression (forward selection, enter limit and remove limit, *p* = 0.10 and *p* = 0.15, respectively) to identify independent predictors of OS and PFS. The survival curves were performed by the Kaplan–Meier method. The log-rank test was used to evaluate the differences between curves. All the experimental methods were performed in accordance with the institutional National and International guidelines and regulations. Statistical evaluation was performed by using the SPSS 29.0 (SPSS Inc., Chicago, IL, USA) for Windows. The dose of the drug that causes 50% cell growth inhibition (IC50) was calculated according to the Chou–Talalay method using the Calcusyn software version 2.11 (Biosoft, Cambridge, UK).

## 5. Conclusions

Overall, the results reported herein show that downregulation of HSP90 expression in preclinical HER2+ breast cancer models in vitro impairs the anti-tumor activity of trastuzumab- and trastuzumab/pertuzumab-based combinations with chemotherapy and hormonal treatment. Interestingly, this does not apply to the activity of T-DM1, which appears to be increased by HSP90 downregulation. From a clinical standpoint, although the prognostic/predictive role of HSP90 expression in HER2+ breast cancer remains ambiguous, further studies are warranted, especially in the settings of hormonal maintenance in patients co-expressing HER2 and ER as well as new HER2-targeted ADCs.

## Figures and Tables

**Figure 1 ijms-26-06593-f001:**
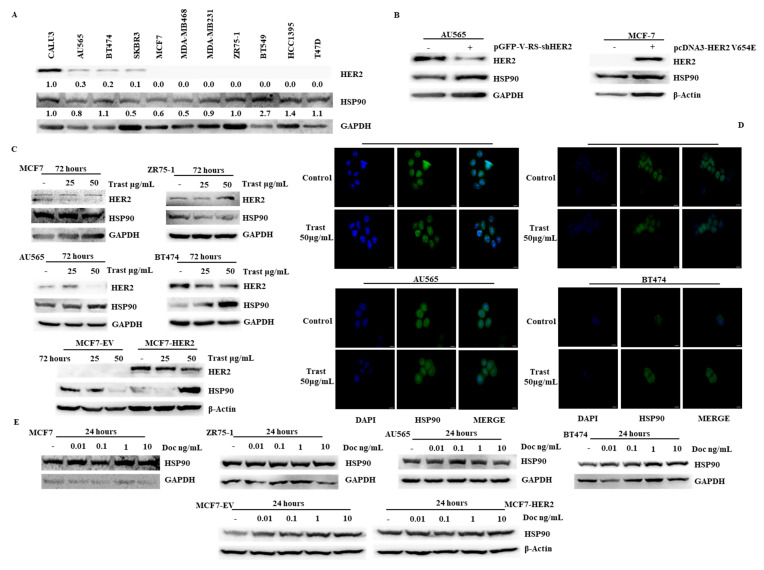
Relationship of HER2 and HSP90 expressions in breast cancer cell lines. (**A**) Breast cancer cell lines, with or without HER2 amplification, were lysed and analyzed by WB using specific antibodies for HER2 and HSP90. WB with a specific antibody for GAPDH is considered as protein loading and blotting control. The CALU3 lung adenocarcinoma cells were used as a positive control for HER2 expression. (**B**) AU565 cells were transfected with a shRNA against HER2 and the result of the transfection was evaluated by WB using specific antibodies for HER2 and HSP90. GAPDH was used as blotting control. MCF-7 cells were transfected with empty vector (MCF7-EV) or with HER2 plasmid constitutively active (MCF7-HER2), and the result of the transfection was evaluated by WB using specific antibodies for HER2 and HSP90. β-Actin was used as blotting control. (**C**–**E**) MCF7, ZR75-1, AU565, and BT474 cells were treated with trastuzumab (Trast) and docetaxel (Doc), at indicated concentrations for 72 and 24 h, respectively. Cell lysates were analyzed by WB using specific antibodies for HER2 and HSP90. Β-Actin and GAPDH were used as blotting control. HSP90 levels, after trastuzumab (Trast) treatment, were also assessed by immunofluorescence assay. Cells were immunostained with anti-HSP90 (green) and the nuclei were stained with DAPI (blue). Microphotographs were taken at ×60 magnification, scale bars 10 µm. Results from one experiment representative of at least three independent experiments are shown.

**Figure 2 ijms-26-06593-f002:**
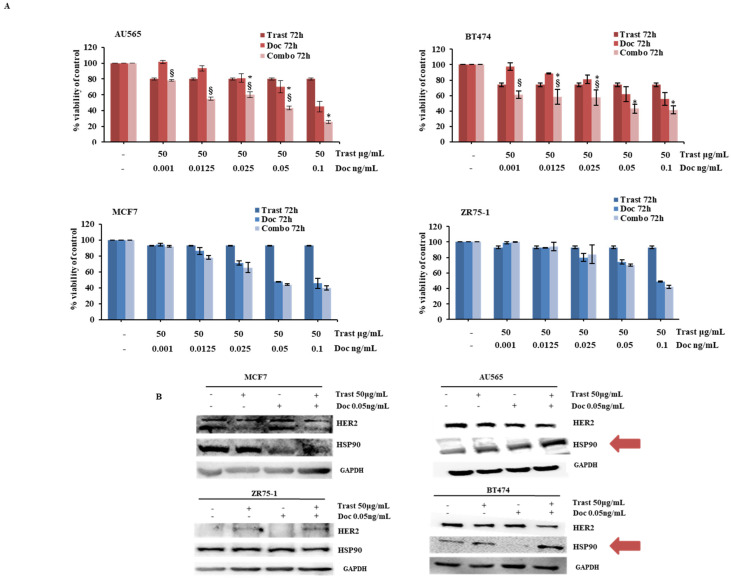
Trastuzumab/docetaxel combination effect in a HER2-driven breast context. (**A**) AU565, BT474, MCF7, and ZR75-1 cells were treated with trastuzumab (Trast) and docetaxel (Doc), alone or in combination (Combo), for 72 h at indicated concentrations. Cell viability was assessed by Crystal Violet assay and results, expressed as percentage of cell growth relative to untreated control, represent the average ±SEM of three independent experiments. Asterisks and symbols indicate statistically significant differences (*p* < 0.05 by 2-tailed Student’s *t* test) for the comparison between * trastuzumab- and combination-treated cells or ^§^ docetaxel- and combination-treated cells. (**B**) MCF7, ZR75-1, AU565, and BT474 cells were treated with trastuzumab (Trast) and docetaxel (Doc), alone or in combination (Combo), for 72 h at indicated concentrations. Cells lysates were analyzed by WB using specific antibodies for HER2 and HSP90 with GAPDH as blotting control.

**Figure 3 ijms-26-06593-f003:**
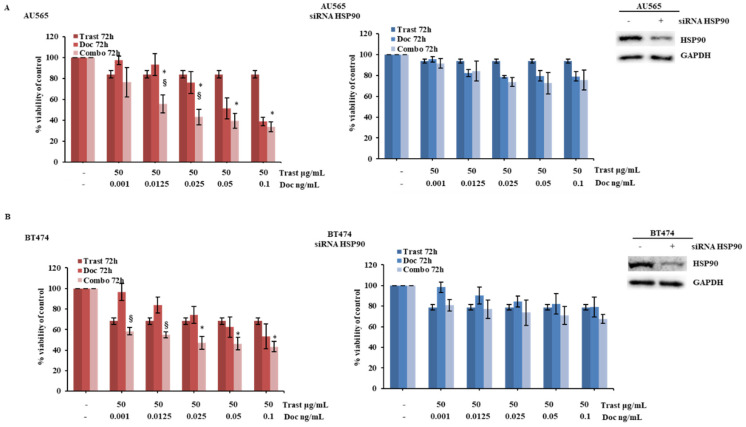
Genetic manipulation of HSP90 expression modifies response to trastuzumab and docetaxel combination. (**A**,**B**) AU565 and BT474 cells were transiently transfected with a siRNA targeting human HSP90 for 72 h, according to the manufacturer’s protocol. HSP90 levels were evaluated by WB using a specific antibody, and GAPDH was used as blotting control. Subsequently, cells were treated with trastuzumab (Trast) and docetaxel (Doc), alone or in combination (Combo), for 72 h at indicated concentrations. Cell viability was assessed by Crystal Violet assay and results, expressed as percentage of cell growth relative to untreated control, represent the average ± SEM of three independent experiments. Asterisks and symbols indicate statistically significant differences (*p* < 0.05 by 2-tailed Student’s *t* test) for the comparison between * trastuzumab- and combination-treated cells or ^§^ docetaxel- and combination-treated cells.

**Figure 4 ijms-26-06593-f004:**
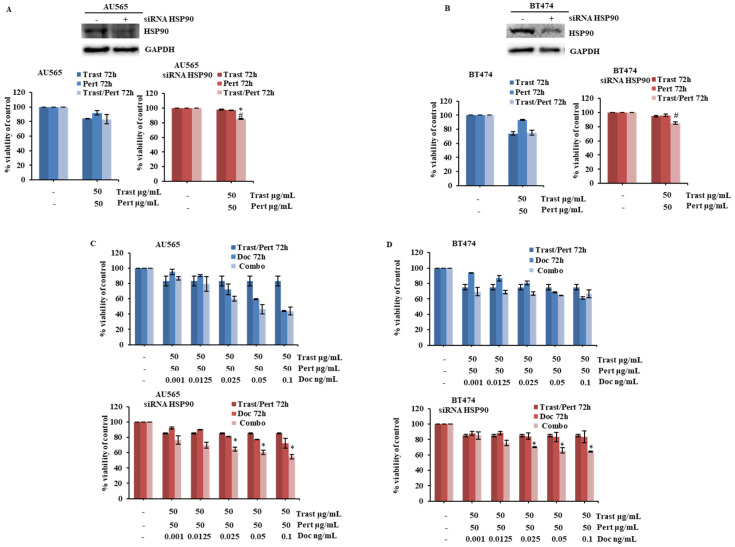
HSP90 downregulation influences the response to the trastuzumab, pertuzumab, and docetaxel combination. (**A**,**B**) AU565 and BT474 cells were transiently transfected with a siRNA targeting human HSP90 for 72 h, according to the manufacturer’s protocol. HSP90 levels were evaluated by WB using specific antibodies for HSP90, and GAPDH was used as blotting control. Subsequently, cells were treated with trastuzumab (Trast) and pertuzumab (Pert), alone or in combination (Trast/Pert), for 72 h at indicated concentrations. Cell viability was assessed by Crystal Violet assay and results, expressed as percentage of cell growth relative to untreated control, represent the average ±SEM of three independent experiments. Asterisks and symbols indicate statistically significant differences (*p* < 0.05 by 2-tailed Student’s *t* test) for the comparison between * trastuzumab- and combination-treated cells or ^#^ pertuzumab- and combination-treated cells. (**C**,**D**) Cells, described previously, were treated with a trastuzumab and pertuzumab combination (Trast/Pert), alone or in combination with docetaxel (Doc), for 72 h at indicated concentrations. Cell viability was assessed by Crystal Violet assay and results, expressed as percentage of cell growth relative to untreated control, represent the average ±SEM of three independent experiments. Asterisks indicate statistically significant differences (*p* < 0.05 by 2-tailed Student’s *t* test) for the comparison between * trastuzumab/pertuzumab- and combination-treated cells.

**Figure 5 ijms-26-06593-f005:**
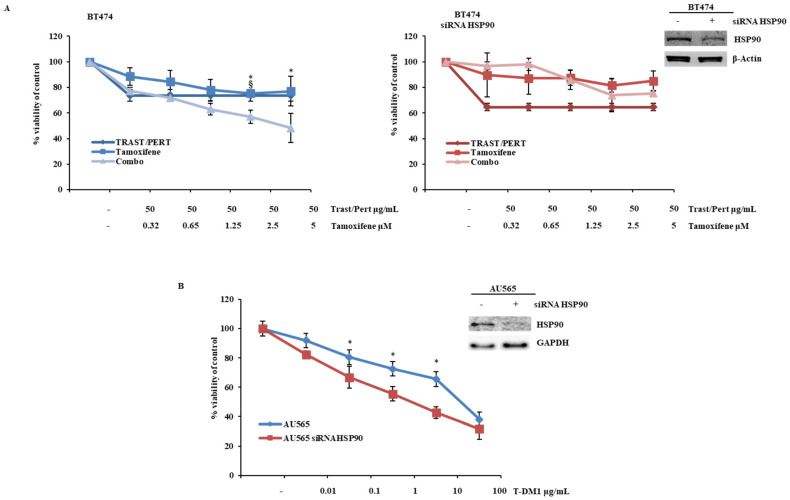
HSP90 silencing influences the response to trastuzumab/pertuzumab/tamoxifen combination and TDM-1 treatment (**A**) BT474 cells were transiently transfected with a siRNA targeting human HSP90 for 72 h, according to the manufacturer’s protocol. HSP90 levels were evaluated by WB using specific antibodies for HSP90, and β-Actin was used as blotting control. Subsequently, cells were treated with trastuzumab/pertuzumab (Trast/Pert) and tamoxifen, alone or in combination (Combo), for 72 h at indicated concentrations. Cell viability was assessed by Crystal Violet assay and results, expressed as percentage of cell growth relative to untreated control, represent the average ±SEM of three independent experiments. Asterisks and symbols indicate statistically significant differences (*p* < 0.05 by 2-tailed Student’s *t* test) for the comparison between * trastuzumab/pertuzumab- and combination-treated cells or ^§^ tamoxifen- and combination-treated cells. (**B**) AU565 cells were transiently transfected with a siRNA targeting human HSP90 for 72 h, according to the manufacturer’s protocol. HSP90 levels were evaluated by WB using specific antibodies for HSP90, and GAPDH was used as blotting control. Subsequently, the cells were treated with trastuzumab emtansine (T-DM1) for 72 h at indicated concentrations. Cell viability was assessed by Crystal Violet assay and the results, expressed as percentage of cell growth relative to untreated control, represent the average ±SEM of three independent experiments. Asterisks indicate statistically significant differences (*p* < 0.05 by 2-tailed Student’s *t* test) for the comparison between * AU565 and AU565 siRNA HSP90.

**Figure 6 ijms-26-06593-f006:**
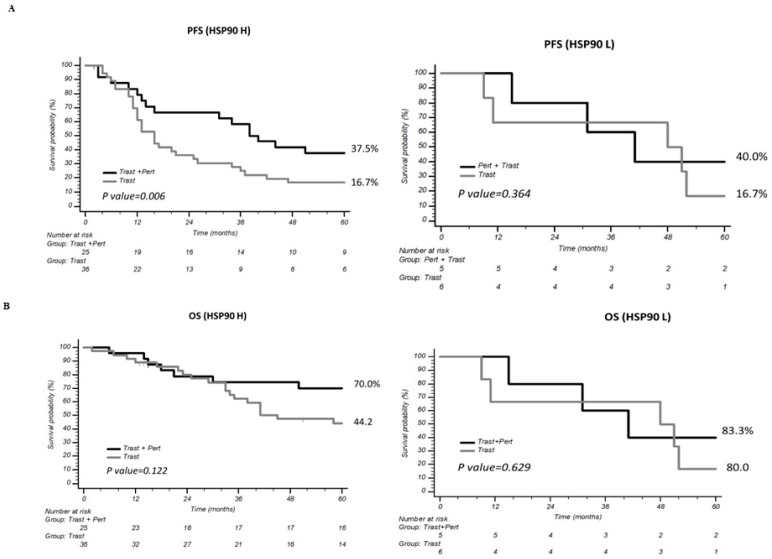
(**A**) Progression-free survival (PFS) with the combination of trastuzumab and pertuzumab, compared to trastuzumab monotherapy, in patients with high and low HSP90 expression. (**B**) Overall survival (OS) with the combination of trastuzumab and pertuzumab, compared to trastuzumab monotherapy, in patients with high and low HSP90 expression.

**Table 1 ijms-26-06593-t001:** Patients’ characteristics.

		Median	Range
Age at metastatic disease		60	37–84
		N	%
Metastases at diagnosis			
	Yes	24	33
	No	48	67
ECOG PS to the first-line treatment			
	0	37	51
	1	25	35
	2	10	14
Histologic specimen			
	Primary tumor (biopsy/surgical specimen)	40	56
	metastatic site	32	44
Neoadjuvant therapy			
	Chemotherapy	3	4
	Chemotherapy + Trastuzumab	12	17
	No	57	79
Surgery			
	Yes	52	72
	No	20	28
Adjuvant therapy			
	Chemotherapy	31	43
	Trastuzumab	23	32
	No	41	57
Estrogen receptors: ERs			
	pos	49	69
	neg	19	27
	n.a.	3	4
Progesteron receptors: PGRs			
	pos	40	56
	neg	28	40
	n.a.	3	4
HER2 status			
	Pos	49	68
	Neg	11	15
	n.a.		
HSP90 expression			
	<250	11	15
	≥250	61	85
ki67			
	≤20	35	49
	>20	33	46
	n.a.	4	5
Grading			
	2	18	25
	3	35	49
	n.a.	19	26
Visceral metastasis			
	Yes	45	63
	No	27	37
	Bone-only disease		
	Yes	12	17
	No	60	83
N° metastatic sites			
	1	35	49
	2	15	21
	3	18	25
	4	3	4
	5+	1	1
Anti-HER2 first-line therapy			
	Trastuzumab alone	42	58
	Trastuzumab+ Pertuzumab	30	42
Maintenance hormone therapy			
	yes	42	58
	no	26	36
	n.a.	4	5

Neg = negative (for HER2 score 0, 1+, or 2+ non-amplified FISH at diagnosis). Pos = positive (for HER2 score 3+). N.A. = not assessed. ECOG PS = ECOG Performance Status. Protein expression was positive in 72 patients (100% of patients). A score from 0 to 3 was assigned for staining intensity, and the percentage of HSP90-positive neoplastic cells was evaluated. To take into account the variability of individual cases both in terms of staining intensity for HSP90 and the percentage of positive neoplastic cells, a hybrid variable was created, obtained by multiplying the intensity of expression (0–3) by the percentage of positive cells (0–100), which was then analyzed to evaluate the possible association between HSP90 expression and clinical variables. Using this method, 61 patients had a positivite score greater than or equal to 250, with only 11 patients having a score < 250; cut-off was calculated by preliminary analysis of the PFS curves.

**Table 2 ijms-26-06593-t002:** Uni- and multivariate analysis for PFS from start of first-line treatment.

Variable	Comparison	*Univariate*			*Multivariate*		
		HR	95% CI	*p*-Value	HR	95% CI	*p*-Value
HSP90 expression	High vs. Low	1.342	0.675–2.665	0.401	-	-	-
Age	Continuous variable	1.017	0.997–1.037	0.089	-	-	-
** *First-line Treatment* **	** *Trastuzumab vs. Trastuzumab + Pertuzumab* **	** *2.178* **	** *1.259–3.769* **	** *0.005* **	** *1.939* **	** *1.105–7.840* **	** *0.021* **
HT Maintenance	Yes vs. No	1.311	0.757–2.270	0.334	-	-	-
Histology	Ductal vs. Lobular vs. Other	-	–	0.531	-	-	-
Surgery	No vs. Yes	1.201	0.687–2.100	0.521	-	-	-
Metastatic de novo	Yes vs. No	1.084	0.631–1.862	0.771	-	-	-
Grading	G3 vs. G2	1.077	0.577–2.010	0.816	-	-	-
ER	Pos vs. Neg	1.570	0.854–2.889	0.147	-	-	-
PGR	Pos vs. Neg	1.254	0.726–2.165	0.416	-	-	-
Ki67	≥20 vs. <20	1.367	0.806–2.317	0.246	-	-	-
Limphnode Met	No vs. Yes	1.174	0.699–1.971	0.545	-	-	-
Visceral Met	No vs. Yes	1.365	0.808–2.308	0.245	-	-	-
Bone-only Disease	Yes vs. No	1.827	0.937–3.564	0.077	-	-	-
CNS Mets	No vs. Yes	1.826	0.569–5.857	0.311	-	-	-
** *ECOG PS* **	** *2 vs. 0/1* **	** *4.383* **	** *2.045–9.393* **	** *<0.001* **	** *3.618* **	** *1.670–7.840* **	** *0.001* **

HT: hormone treatment; Met/Mets: metastases; CNS: central nervous system; ECOG PS: Performance Status according to the Eastern Cooperative Oncology Group; Lymph node Met = metastatic lynphonodes; Visceral Met = visceral metastases; CNS Mets = central nervous system metastases.

## Data Availability

Further information related to the current study is available from the corresponding author upon reasonable request.

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
