# Peer review of "HER2-Driven Breast Cancer: Role of the Chaperonin HSP90 in Modulating Response to Trastuzumab-Based Therapeutic Combinations"

_ijms, 2025, doi:10.3390/ijms26146593_

Round 1
Reviewer 1 Report
Comments and Suggestions for Authors
The article New Therapeutic Approaches in HER2-Driven Breast Cancer: Role of the Chaperonin HSP90 in Response to Pharmacological Treatments is an interesting article that studies the relation between HER-2 and HSP90 proteins when different cells are treated with Trastuzumab, Pertuzumab y docetaxel. I have some suggestion before the article could be accepted for publication.
1.- Despite that is mentioned that in normal conditions there are not a correlation between these two proteins could be that the correlation exist under some specific conditions for example that when when HER is diminish HSP90 increased
2.- Could be possible mention in the title the drugs evaluated
3.- Please revise the letter labels in the figure 1
4.- In the figure 2 instead to use doses employ concentrations due to is for in vitro experiments no in vivo
5.- HSP90 expression has been reported in all cells evaluated?
6.- Figure 3 the label in the exe y are not % of control is % of viability, obtained considering at the control as 100%, or why it was expressed as % of control
7.- The title for the subtitle 2.5 should by in cursive
8.- In the table 1 the HSP90 in the patients mention mayor to 250 but which are the units?? In the methodology mention the immunohistochemistry for HSP90 in the tissue samples but any image about this is shown.
9.- Why the cell viability was measured at 72 and 96h and not at 24 or 48 h.
10.- In the discussion there are parts where the reference is missing “success so far (ref)” and “n and degradation (ref)”
11.- The conclusion sound confusing due to in the discussion section the authors mention that “The set of experiments presented here was designed to assess the functional relevance of HSP90 expression in the response to HER2-targeted therapy, rather than HER2 dynamics and signaling in relationship to HSP90 modulation; in that respect our data cannot contribute to settle the question of whether high HSP90 expression fosters or restrains HER2 signaling. From a functional standpoint, however, HSP90 downregulation by genetic means clearly blunts the response to combined trastuzumab/docetaxel, thereby abrogating their therapeutic synergism. Then, I suggest to re write the conclusion focused on the use of the treatments and their combination.
Author Response
Dear Editor,
enclosed please find a thoroughly revised version of the manuscript entitled "HER2-Driven Breast Cancer: Role of the Chaperonin HSP90 in Modulating Response to Trastuzumab-based Therapeutic Combinations", which we would like to resubmit to International Journal of Molecular Sciences. Please note that, following the suggestion of reviewer #1 (comment #2, see below), the original title (New therapeutic approaches in HER2-driven breast cancer: role of the chaperonin HSP90 in response to pharmacological treatments) has been modified to mention the drugs evaluated.
We would like to thank the reviewers for their accurate analysis of the manuscript and for highlighting important critical points. We have extensively revised the manuscript accordingly and believe this has improved the manuscript's quality, so that it can now be published in your prestigious Journal.
Reviewer 1.
- Despite that is mentioned that in normal conditions there are not a correlation between these two proteins could be that the correlation exist under some specific conditions for example that when when HER is diminish HSP90 increased.
Response: Although no significant correlation between HER2 and HSP90 expression exists under basal conditions in breast cancer cell lines or in published literature of breast cancer patient series, we show in Figure 1B-C that the genetic or pharmacological downregulation of HER2 results in increased expression of HSP90; conversely, we do not observe substantial changes in HSP90 expression, after genetic upregulation of HER2. This is extensively addressed in the Discussion section (lines 590-608).
- Could be possible mention in the title the drugs evaluated
Response: as suggested, we modified the title into: “HER2-Driven Breast Cancer: Role of the Chaperonin HSP90 in Modulating Response to Trastuzumab-based Therapeutic Combinations”.
- Please revise the letter labels in the figure 1
Response: In the Figure 1, the panel labels have been modified to improve the figure comprehension
- In the figure 2 instead to use doses employ concentrations due to is for in vitro experiments no in vivo
Response: as requested, in legend of Figure 2 we modified the text in “concentrations” and this change is reported in all figures and legends of the revised manuscript.
- HSP90 expression has been reported in all cells evaluated?
Response: yes, basal expression of HSP90 is evaluated in all cell contexts considered in this manuscript (Figure 1A). Moreover, its modulation (due to response to pharmacological treatment or genetic manipulation) is described in all Figures, where necessary.
- Figure 3 the label in the exe y are not % of control is % of viability, obtained considering at the control as 100%, or why it was expressed as % of control
Response: as correctly requested, we have modified the y-axe in % viability of control. This change is applied to all manuscript figures.
- The title for the subtitle 2.5 should by in cursive
Response: the title font is changed in the text
- In the Table 1 the HSP90 in the patients mention mayor to 250 but which are the units?? In the methodology mention the immunohistochemistry for HSP90 in the tissue samples but any image about this is shown.
Response: for each patient's sample, the percentage of positive neoplastic cells is multiplied by staining intensity: 1 (weak), 2 (moderate), 3 (strong). Example: 50% neoplastic cells * staining 2= 100. This method is described in the legend to Table 1 and in the Results section (lines 456-462). Examples of immunohistochemistry image are now shown in Figure S6.
- Why the cell viability was measured at 72 and 96h and not at 24 or 48 h.
Response: cell viability experiments were also conducted at 24 and 48 hours of treatment. However, at such time points no significant impact on cellular growth inhibition were observed especially for treatment with trastuzumab, docetaxel, and their combination. In addition, the effects of trastuzumab showed little differences, in terms of cellular growth inhibition, between 24-48 and 72-96 hours. Below, we add two examples of the experiments conducted at 48hours on AU565 and BT474 context (see attached file).
- In the discussion there are parts where the reference is missing “success so far (ref)” and “n and degradation (ref)”
Response: we apologize for the oversight. We have added the missing references
- The conclusion sound confusing due to in the discussion section the authors mention that “The set of experiments presented here was designed to assess the functional relevance of HSP90 expression in the response to HER2-targeted therapy, rather than HER2 dynamics and signaling in relationship to HSP90 modulation; in that respect our data cannot contribute to settle the question of whether high HSP90 expression fosters or restrains HER2 signaling. From a functional standpoint, however, HSP90 downregulation by genetic means clearly blunts the response to combined trastuzumab/docetaxel, thereby abrogating their therapeutic synergism. Then, I suggest to re write the conclusion focused on the use of the treatments and their combination.
Response: We thank the reviewer for his/her suggestion; the Conclusions section has been modified to better highlight the impact of HSP90 expression of response to the different treatment strategies (lines 831-838).

Reviewer 2 Report
Comments and Suggestions for Authors
In the present original research paper, the authors investigate the role of HSP90 expression in the response of breast cancer cells to HER2-targeted treatments, including cell line models and samples from metastatic, HER2+, breast cancer patients who have received trastuzumab- or trastuzumab + pertuzumab-based first-line treatment. The cell line models include HER2-positive cell lines and in some experiments also cell lines where the HER2 expression has been induced or downregulated by transfection, which together with mock-treated cells present excellent controls for the observations. Tested HER2-targeting drugs include trastuzumab, docetaxel, their combination, but also the antibody-drug conjugate T-DM1. Methodically, the HER2 and HSP90 expression is determined using Western blotting, and cytotoxicity is measured using proliferation tests with crystal violet. In these experiments, it was found that HER2 downregulation achieved by transfection or trastuzumab treatment induced HSP90 upregulation and potentiated synergistic effect on growth inhibition of trastuzumab and docetaxel combination. HSP90 downregulation decreased the response to trastuzumab and docetaxel and their synergism.
In further experiments, HSP90 modulation was studies in response to the combination of trastuzumab, pertuzumab and tamoxifen in the BT474 cell line. HSP90 silencing reduced the growth inhibitory response to tamoxifen alone and inhibited the anti-tumor effect of the combination. On the contrary, the growth inhibition by the ADC T-DM1 in the AU565 cell line model was increased when HSP90 expression was silenced.
For patients’ samples HSP90 expression was also assessed by immunohistochemistry in a series of 72 metastatic HER2 positive breast cancer patients. The increased efficacy of trastuzumab/pertuzumab combination vs. trastuzumab alone treatment, when expressed in progression-free survival upon hormonal maintenance, was mostly typical for the group of patients with high HSP90. Otherwise, a significant correlation between the HSP90 level and the outcome depending on the type of anti-HER2 treatment (trastuzumab/pertuzumab vs trastuzumab alone) could not be demonstrated.
This is a very valuable manuscript containing several preclinical data, which are of potential translational value and could improve HER2-directed targeted therapy design, when considering the examination of HSP90 level as an additional marker. Data are well analyzed and presented, text is interestingly written and conclusions critically discussed. Outlook is also very interesting, especially regarding future studies into HSP90 level relation with the success of anti-HER2 ADC therapy.
Please find below a list of remarks which I hope will be helpful.
Line 55: Tra/Per combinations, should this be combination? Please also correct throughout the text.
Figure 1. GAPDH should be capitalized, and in the Figure 1E the WB for AU565 cells says “Gapgh) – please correct. Figure 1D, microscopy: micron bar is missing
Figure 2: the y-axes should be labeled “% viability of control”. Please correct in all graphs including those in the supplementary material.
Line 154: the presentation of statistical significance for different conditions with multiple asterisks is not that well chosen because these usually mean different significance levels – I suggest that other symbols are used.
Line 157: Cell lysates were analyzed
Lines 224-226: should be in subtitle formatting
Line 237: T-DM1 (please correct throughout the manuscript, including the Figures).
Line 250: transiently transfected
Line 250: manufacturer’s protocol
Line 273: the note obviously refers to Her2 status, where it is labeled with neg1, and 1 should point to the note? Table 1 is pasted as a Figure and could be formatted better. Abbreviations require explanation at the Table Title.
Line 280: the sentence should continue?
Table 2: Lymphnode, and Met/Mets should be spelled out
Line 383: (ref) – reference missing
Line 417: (ref) – reference missing
Line 457: tamoxifen growth inhibitory activity
Line 463: CO2, 2 in subscript
Line 472: all commercially acquired antibodies should be listed with the Supplier, Catalogue Number and RRID
Line 647: reference 19: “ATCC. Availabe online: https://www.atcc.org/ (accessed on” – incomplete
Table S1: All cell lines should be cited with RRID identificators and ATCC numbers
Author Response
Dear Editor,
enclosed please find a thoroughly revised version of the manuscript entitled "HER2-Driven Breast Cancer: Role of the Chaperonin HSP90 in Modulating Response to Trastuzumab-based Therapeutic Combinations", which we would like to resubmit to International Journal of Molecular Sciences. Please note that, following the suggestion of reviewer #1 (comment #2, see below), the original title (New therapeutic approaches in HER2-driven breast cancer: role of the chaperonin HSP90 in response to pharmacological treatments) has been modified to mention the drugs evaluated.
We would like to thank the reviewers for their accurate analysis of the manuscript and for highlighting important critical points. We have extensively revised the manuscript accordingly and believe this has improved the manuscript's quality, so that it can now be published in your prestigious Journal.
Reviewer 2
- Line 55: Tra/Per combinations, should this be combination? Please also correct throughout the text.
- Line 157: Cell lysates were analyzed
- Lines 224-226: should be in subtitle formatting
- Line 237: T-DM1 (please correct throughout the manuscript, including the Figures).
- Line 250: transiently transfected
- Line 250: manufacturer’s protocol
- Line 280: the sentence should continue?
- Line 383: (ref) – reference missing
- Line 417: (ref) – reference missing
- Line 457: tamoxifen growth inhibitory activity
- Line 463: CO2, 2 in subscript
Response: as requested, all changes have been incorporated into the text.
- Figure 1. GAPDH should be capitalized, and in the Figure 1E the WB for AU565 cells says “Gapgh) – please correct. Figure 1D, microscopy: micron bar is missing
Response: GAPDH was capitalized throughout the text and in all figures. As requested, scale bar has been added to the immunofluorescence images.
- Figure 2: the y-axes should be labeled “% viability of control”. Please correct in all graphs including those in the supplementary material.
Response: this change has been made in all figures of manuscript.
- Line 154: the presentation of statistical significance for different conditions with multiple asterisks is not that well chosen because these usually mean different significance levels – I suggest that other symbols are used.
Response: all the figures are modified with other symbols to highlight statistical significance for different conditions
- Line 273: the note obviously refers to Her2 status, where it is labeled with neg1, and 1 should point to the note? Table 1 is pasted as a Figure and could be formatted better. Abbreviations require explanation at the Table Title.
Response: the Table 1 has been reformatted and the abbreviations have been included. Neg 1 has been corrected
- Table 2: Lymphnode, and Met/Mets should be spelled out
Response: Spelling of abbreviations has been added as Table 2 legend (lines 549-550)
- Line 472: all commercially acquired antibodies should be listed with the Supplier, Catalogue Number and RRID
Response: the Catalogue Numbers and RRID ID, for all antibodies used in this study, have been included in the text.
- Line 647: reference 19: “ATCC. Availabe online: https://www.atcc.org/ (accessed on” – incomplete
Response: reference 19 has been updated to the date 20-05-2025
- Table S1: All cell lines should be cited with RRID identificators and ATCC numbers
Response: ATCC numbers and RRID ID have been added for all cell lines included in Table S1.
